# CRISPR-mediated Transfection of *Brugia malayi*

**Canhui Liu[1], Alexandra Grote[2¤], Elodie Ghedin[2,3], Thomas R. Unnasch** [1] *****

**1** Center for Global Health Infectious Disease Research, College of Public Health, University of South Florida, Tampa, Florida, United States of America, **2** Center for Genomics and Systems Biology, Department of Biology, New York University, New York, New York, United States of America, **3** Department of Epidemiology, School of Global Public Health, New York University, New York, New York, United States of America

¤ Current address: Broad Institute of MIT and Harvard, Cambridge, Massachusetts, United States of America
* tunnasch@usf.edu

**Data Availability Statement:** All relevant data are within the manuscript and its Supporting Information files.

**Funding:** This research was supported by a grant from the National Institute of Allergy and Infectious Diseases to TRU (Award 1 R21 AI135172-01A1).

## Abstract

The application of reverse genetics in the human filarial parasites has lagged due to the difficult biology of these organisms. Recently, we developed a co-culture system that permitted the infective larval stage of *Brugia malayi* to be transfected and efficiently develop to fecund adults. This was exploited to develop a *piggyBac* transposon-based toolkit that can be used to produce parasites with transgene sequences stably integrated into the parasite genome. However, the *piggyBac* system has generally been supplanted by Clustered Regularly Interspaced Short Palindromic Repeats (CRISPR) based technology, which allows precise editing of a genome. Here we report adapting the *piggyBac* mediated transfection system of *B. malayi* for CRISPR mediated knock-in insertion into the parasite genome. Suitable CRISPR insertion sites were identified in intergenic regions of the *B. malayi* genome. A dual reporter *piggybac* vector was modified, replacing the *piggyBac* inverted terminal repeat regions with sequences flanking the insertion site. *B. malayi* molting L3 were transfected with a synthetic guide RNA, the modified plasmid and the CAS9 nuclease. The transfected parasites were implanted into gerbils and allowed to develop into adults. Progeny microfilariae were recovered and screened for expression of a secreted luciferase reporter encoded in the plasmid. Approximately 3% of the microfilariae were found to secrete luciferase; all contained the transgenic sequences inserted at the expected location in the parasite genome. Using an adaptor mediated PCR assay, transgenic microfilariae were examined for the presence of off target insertions; no off-target insertions were found. These data demonstrate that CRISPR can be used to modify the genome of *B. malayi*, opening the way to precisely edit the genome of this important human filarial parasite.

## Author summary

Human filarial parasites are the causative agents of lymphatic filariasis (elephantiasis) and onchocerciasis (river blindness) and are some of the most important causes of morbidity worldwide. A large obstacle to research on these organisms has been the inability to

The funders had no role in study design, data collection and analysis, decision to publish, or preparation of the manuscript.

**Competing interests:** The authors have declared that no competing interests exist.

employ reverse genetic methods and to develop integrated transgenic parasite lines. Recently, we developed a *piggyBac* transposon-based method that employed a co-culture system that permitted the infective larval stage of *B. malayi* to be transfected by lipofection in culture, resulting in the production of developmentally competent transgenic parasites. However, the *piggyBac* system cannot be used to precisely edit particular sequences in the genome. Thus, the *piggyBac* system has generally been supplanted by Clustered Regularly Interspaced Short Palindromic Repeats (CRISPR) based technology, which permits precise targeting (and editing) of particular sequences in the genome. Here, we report building upon the methods developed for *piggyBac* mediated transfection of *B. malayi* to develop a CRISPR mediated method for precise transgenesis in this parasite.

## Introduction

The human filarial parasites are the causative agents of two of the most important diseases of mankind in terms of socioeconomic disruption, lymphatic filariasis and onchocerciasis. Both of these diseases are currently the subject of international efforts to achieve elimination [1], primarily through the use of community directed treatments using donated drugs [2, 3]. These programs all rely on the use of a small number of drugs that must be given repeatedly over a long period of time to achieve elimination [4]. The need to obtain high drug coverage over a long period of time is logistically difficult and expensive, and the reliance on a limited number of drugs makes the programs vulnerable to failure if resistance develops to the currently available antifilarial agents [5]. Thus, further research to develop new drugs and to better understand the mechanisms of resistance to the currently available drugs is needed.

Reverse genetic techniques have become essential tools in the effort to develop new drugs and understand the mechanism of drug resistance in many pathogen systems. Unfortunately, the human filarial parasites have lagged behind many other organisms in the development and application of these technologies. This is partially due to the difficult biology of these organisms as they do not survive outside of their vertebrate hosts. Furthermore, only *Brugia malayi* (a parasite causing lymphatic filariasis in Southeast Asia) can be cultured in a variety of surrogate animal hosts, most efficiently in the Mongolian jird or gerbil (*Meriones unguiculatus*) [6–9].

Despite these obstacles, some progress has been made in developing methods to genetically modify the filariae. Recently, we developed a method relying upon a novel co-culture system that permitted the infective larval stage of *B. malayi* to be transfected by lipofection in culture, resulting in the production of developmentally competent transgenic parasites [10]. We exploited this method to develop a number of reporter plasmids based upon the heterologous *piggyBac* transposon-based system that can be used to produce transgenic parasites in which the transgenic sequences are stably integrated into the parasite genome. However, the *piggyBac* system has some inherent disadvantages for studies of gene expression, such as the semi-random nature of the insertion point which can result in effects difficult to control. These may include position effects resulting from insertion into different locations in the genome (euchromatin versus heterochromatin, for example) as well as the possibility of multiple insertions into a single genome, resulting in copy number variation. In addition, the *piggyBac* system cannot be used to precisely edit particular sequences in the genome. Thus, the *piggyBac* system has generally been supplanted by Clustered Regularly Interspaced Short Palindromic Repeats (CRISPR) based technology [11], which permits precise targeting (and editing) of particular sequences in the genome. CRISPR has recently been applied to genetically modify a

number of parasitic worm species (for reviews see [12–14]). Most of the studies that have applied CRISPR to parasitic worms have been conducted using *Strongyloides spp.*, which have the advantage of having free-living stages in their lifecycle. CRISPR has been used in *Strongyloides* to disrupt genes involved in development [12], motility [15] and thermotaxis [16]. CRISPR has also been shown to be a useful tool for producing targeted genetic modifications in the liver fluke *Opisthorchis viverrini* [17] and in the blood fluke *Schistosoma mansoni* [18]. Here, we report building upon the methods developed for *piggyBac* mediated transfection of *B. malayi* to develop a CRISPR mediated method for precise transgenesis in this parasite.

## Materials and methods

### Ethics statement

Protocols involving animals were reviewed by the Institutional Animal Care and Use Committee of the University of South Florida and were approved under protocol # R IS00003568. The Animal Facilities at the University of South Florida are fully accredited by AAALAC International as program #000434 and are managed in accordance with the Guide for the Care and Use of Laboratory Animals, the Animal Welfare Regulations, the PHS Policy, the FDA Good Laboratory Practices, and the IACUC Principles and Procedures of Animal Care and Use. The University of South Florida has assurance #D16-00589 (A4100-01) on file with OLAW/PHS and maintains registration #58-R-0015 with USDA/APHIS/AC.

### Preparation of pCHR1-BmGLuc-GFP

pCHR1-BmGLuc-GFP was derived from two plasmids, pBACII-EGDH and pBACIIBmGLuc-GFP [10]. As a first step in this process, the plasmid backbone was amplified from pBACII-EGDH. The amplification targeted positions 4142–726 on the published sequence, which eliminated the ITRs and GFP expression cassette from pBACII-EGDH. The primers used in the amplification were:

BACIIF': 5' GGACTTTGTCCAGGATCATATCGTCGGGTCTT 3'

BACIIR': 5' GGGGCCCGTTCTGCAGCGTGTCGAGCAT 3'

BACIIF' contains a synthetic *Pfl* F1 site and BACIIR' contains a synthetic *Apa* 1 site for subsequent cloning. These are underlined in the sequences.

Amplification reactions used Thermo Scientific Maxima Hot Start PCR Master Mix (Thermo Fisher, Harrisburg, PA). Amplification reactions were carried out using 100 ng of pBACll-EGDH DNA as a template. Cycling conditions consisted of an initial denaturation step at 94˚C for 5min, followed by 5 cycles consisting of 94˚C for 30 sec, 52˚C for 30 sec and 72˚C for 5min, followed by 25 cycles consisting of 94˚C for 30 sec, 57˚C for 30 sec and 72˚C for 5 min. The reaction was completed by an incubation at 72˚C for 10 min. The resulting amplicon was gel-purified on a 1% agarose gel and the 5' ends of the amplicon phosphorylated with T4 DNA kinase (New England Biolabs, Ipswich, MA) following the manufacturer's instructions. The blunt ends of the amplicon were self-ligated and the resulting circular DNA recovered by transformation into One Shot TOP10 chemically competent *E. coli* cells (Thermo Fisher) to produce an empty plasmid backbone.

The Bm*hsp70*-GLuc Bm*rps12*-GFP expression cassette was then amplified from pBACIIBmGLuc-GFP using the following primers:

GFPGLUC F' 5' GGGGCCCCTAGAACAATATTCACAAGGAC 3'

GFPGLUC R' 5' GGACTTTGTC GCCCAAGCAATTTCGAATGAAGGA 3'

GFPGLUC F' contains a synthetic *Apa* 1 site and GFPGLUC R' contains a synthetic *Pfl* F1 site. These sites are underlined in the sequences. Amplification reactions were carried out using Thermo Scientific Maxima Hot Start PCR Master Mix with 100 ng of pBACII-BmGLuc-GFP DNA as a template. Cycling conditions consisted of an initial denaturation step at 94˚C for 5min, followed by 20 cycles consisting of 94˚C for 30 sec, 52˚C for 30 sec and 72 $^O$C for 5min. The reaction was completed with a final extension at 72 $^O$C for 10 min. The resulting amplicon was cloned into the PCRII cloning vector (Thermo Fisher) and the sequence of the amplicon confirmed.

Both the vector and the TA clone were then digested with *Pfl* F1 and *Apa* 1. The insert from the PCRII cloning vector was gel-purified on a 1% agarose gel, and the vector was treated with alkaline phosphatase. The vector and the insert were then ligated and plasmids recovered following transformation into One Shot TOP10 chemically competent *E. coli* cells (Thermo Fisher).

The downstream domain of the insertion site was then amplified from *B. malayi* genomic DNA with the following primers:

BmCHR1 c6485129: 5' GGACTTTGTCCaaatgattgtattggatagaataatgcga 3'

BmCHR1 nc6485987: 5' GGACTTTGTCCTTGTGACGCCACTTGTATTGCT 3'

The primers included synthetic *Pfl* F1 sites (underlined) for subsequent cloning. Amplification reactions were carried out using Thermo Scientific Maxima Hot Start PCR Master Mix with 100 ng of *B. malayi* genomic DNA as a template. Cycling conditions consisted of an initial denaturation step at 94˚C for 5min, followed by 35 cycles consisting of 94˚C for 30 sec, 52˚C for 30 sec and 72˚C for 2min. The reaction was completed with a final extension at 72˚C for 10 min. The resulting amplicon (876 bp including the extra nucleotides at the 5' ends of the primers) was cloned into the PCRII cloning vector and its sequence confirmed. The plasmid containing the expression cassette and the TA clone with the downstream domain were then digested with *Pfl* F1. The downstream domain was gel-purified on a 1% agarose gel and the digested vector treated with alkaline phosphatase as described above. The insert and plasmid fragments were ligated and recovered following transformation into One Shot TOP10 chemically competent *E. coli* cells. The identity and orientation of the insert in the resulting plasmid was confirmed by DNA sequencing.

The upstream domain of the insertion site was amplified from *B. malayi* genomic DNA using the following primers:

BmCHR1 c6484000: 5' GGGGCCCGcaactacattgagaattgttgta 3'

BmCHR1 nc6485105: 5' GGGGCCCTgaacggaaaacacaaaaacttataa 3'

The primers included synthetic *Apa* 1 sites (underlined) for subsequent cloning. Amplification reactions were carried out using Thermo Scientific Maxima Hot Start PCR Master Mix with 100 ng of *B. malayi* genomic DNA as a template. Cycling conditions consisted of an initial denaturation step at 94˚C for 5min, followed by 35 cycles consisting of 94˚C for 30 sec, 52˚C for 30 sec and 72˚C for 2min. The reaction was completed with a final extension at 72˚C for 10 min. The resulting amplicon (1118 bp including the extra nucleotides from the 5' ends of the primers) was cloned into the PCRII cloning vector and its sequence confirmed. The plasmid containing the expression cassette and the downstream insertion domain and the TA clone with the upstream domain were then digested with *Apal* 1. The upstream domain fragment was gel-purified on a 1% agarose gel and the digested vector treated with alkaline phosphatase as described above. The insert and plasmid fragments were ligated and recovered following transformation into One Shot TOP10 chemically competent *E. coli* cells. The identity and

orientation of the insert in the resulting plasmid was confirmed by sequencing. The activity of the expression cassette in this final construct (designated pCHR1-BmGLuc-GFP) was confirmed by biolistic transfection of isolated *B. malayi* embryos followed by assaying the culture medium for secreted GLuc activity, as previously described [19].

## Preparation of sgRNA

Chr1scaffold 001:6484000–6485999 was analyzed to identify the best target location for CRISPR cutting and insertion using the target selection program ChopChop (chopchop.rc.fas. harvard.edu). The best target sequence was located at positions corresponding to the sequence 5' TTTGATTTACTATCTCTTCT<u>CGG</u> 3'. The PAM sequence (NGG, underlined), while required for Cas9 recognition of the target sequence, is not part of the sgRNA sequence. A terminal G was then appended to the 5' end of the sequence to promote *in vitro* transcription, and the sequence of the T7 promoter was then added to the 5' end. A 14 nucleotide overlap sequence was then appended to the 3' end of the oligonucleotide. This resulted in the final oligonucleotide sequence:

5' ttctaatacgactcactata**g**TTTGATTTACTATCTCTTCT<u>gttttagagctaga</u> 3'.

The sequence of the final sgRNA is indicated in uppercase, the T7 promoter sequence in regular lowercase, the single G added to promote transcription in bold and the 3' overlap in underlined lowercase. This oligonucleotide was synthesized by a commercial vendor (Eurofin Genomics, Louisville, KY).

sgRNAs were generated from the synthesized oligonucleotide using the EnGen sgRNA Synthesis Kit, *S. pyogenes* (New England Biolabs) following the manufacturer's instructions. The resulting sgRNA was purified using the Monarch RNA Cleanup Kit (New England Biolabs). The purified sgRNA was analyzed for integrity and purity by 15% polyacrylamide gel electrophoresis.

## *In vitro* confirmation of sgRNA directed cutting of the target sequence

The genomic sequence corresponding to the upstream and downstream domains flanking the insertion site were amplified using 100 ng of *B. malayi* genomic DNA as a template. The PCR was performed using Thermo Maxima DreamTaq Hot Start PCR with the primers BmCHR1 c6484000 and BmCHR1 nc6485987. The cycling conditions consisted of 4 min at 94˚C followed by 35 cycles consisting of 30 sec at 94˚C, 30 sec at 55˚C and 2 min at 72˚C. The reaction was completed with a final extension for 10 min at 72˚C. The resulting amplicon was purified on a 1% agarose gel.

A solution containing 3 μl of 10X NEBuffer 3.1 (New England Biolabs), 3 μl of 300nM sg RNA, and 1 μl of 1uM EnGen Cas9 Nuclease NLS (New England Biolabs) in a final volume of 27 μl was prepared and incubated at 25˚C for 10 minutes. A total of 3 μl of a 30nM solution of the amplicon from the above reaction was then added, bringing the final volume to 30 μl. The reaction was incubated at 37˚C for 15 minutes. A total of 1 μl of Proteinase K (0.8 units; New England Biolabs) was added to stop the reaction, and the mixture incubated at room temperature for an additional 10 minutes. The resulting reaction products were then analyzed on a 1% agarose gel.

## Lipofection, implantation and recovery of *B. malayi*

*B. malayi* infective larvae (L3) were obtained from the Filarial Research Reagent Resource Center (FR3). Prior to the arrival of the L3, the individual wells of a 24-well tissue culture plate were seeded with 1x10$^5$ Bovine Embryo Skeletal Muscle (BESM) cells/well. The cells were

cultured for 1–2 days in Minimal Essential Media (MEM) containing 20% fetal bovine serum (FBS) until they reached 70% to 90% confluency. Upon receipt, the L3 were washed five times with a solution consisting of RPMI 1640 medium containing 0.1x Antibiotic Antimycotic solution (Gibco), 10 µg/ml gentamycin and 2 µg/ml Ciprofloxin. A total of 200 L3 were then dispersed in 5 ml of RPMI 1640 medium containing 25 mM HEPES, 20% fetal calf serum, 20 mM glucose, 24 mM sodium bicarbonate, 2.5 µg/ml amphotericin B, 100 U/ml penicillin, 100 U/ml streptomycin, 40 µg/ml gentamicin, 2 µg/ml Ciprofloxin and 2 µg/ml Fortaz (CF-RPMI). The L3 were allowed to settle and all but 1ml of medium removed. The medium on the feeder cells was replaced with CF-RPMI and transwells (Costar, 3.0 um pore size) were placed in the wells. The dispersed L3 were then aliquoted among the transwells so that each transwell contained approximately 100 L3. Additional CF-RPMI was added to the well to bring the total volume in the well to 1mL.

Lipofection of the cultured larvae used two micelle preparations, one containing the CAS9-RNP complexes and one containing the plasmid DNA. The CAS9-RNP micelles were prepared by diluting 18 µl RNAiMAX reagent (Thermo Fisher) in 300 µl Opti-MEM medium (Thermo Fisher). A total of 60 pmol of sgRNA (2 µg) and 60pmol of EnGen Cas9 Nuclease NLS (New England Biolabs) were then added to 300 µl Opti-MEM medium. This was gently mixed and incubated at room temperature for 10 minutes to form the CAS9- RNP micelles. The RNP complexes were then combined with 300 µl of the diluted RNAiMAX reagent and the mixture was incubated for 20 minutes at room temperature. A total of 50µl of the Cas9-RNP liposome complex solution was then added to each transwell containing the L3.

The plasmid micelles were prepared by diluting 24 µl Lipofectamine LTX reagent (Thermo Fisher) in 300 µl Opti-MEM medium (Thermo Fisher). A total of 6 µg of pCHR1-BmGLuc-GFP plasmid DNA were added to 300 µl Opti-MEM medium; 6 µl PLUS Reagent was then added to the DNA solution. This solution was combined with 300 µl of the diluted Lipofectamine LTX Reagent to prepare the DNA micelles. The mixture was incubated for 5 minutes at room temperature. A total of 50 µl of the micelle solution was then added to each well containing the L3 and the CAS9-RNP liposome complexes. The L3 were then cultured at 37˚C under 5% $CO_2$. The feeder cell medium was changed and additional freshly prepared micelle solutions were added to the transwells on a daily basis. On day 5, molting of the L3 was induced by the inclusion of ascorbic acid to a final concentration of 75 µM in the feeder cell medium. The L3 were incubated for a total of eight days. On day 8 (three days after molting was induced) the transfected larvae were collected and injected intraperitoneally into two naive jirds using an 18-gauge needle. Approximately 100 L3 were injected into each animal. The parasites were allowed to develop for 140 days. The animals were then euthanized and microfilariae recovered from the peritoneal cavity. Recovered microfilariae were cultured individually in 100 µl of CF-RPMI for 48 hours in 96 well plates and the medium assayed for secreted GLuc activity as previously described [19].

### Hemi-nested PCR analysis of transgene insertion sites

Individual microfilaria were placed in 100 µl of nuclease-free deionized water and subjected to two freeze-thaw cycles. The samples were then heated to 100˚C for 10 minutes and 1 µl of proteinase K (1mg/ml) was added. The samples were incubated at 55˚C for 50 minutes and the proteinase K inactivated by heating the samples to 95˚C for 10 minutes.

Insertion sites were amplified from the DNA isolated from the individual parasites using a hemi-nested PCR protocol. Hemi-nested PCR was performed using Thermo Maxima Dream-Taq Hot Start PCR master mix with 5 µl of the solution described above as a template and the following primers:

RPS12 UTR F: 5' GACTATTGTTTGTTTATTGTTTGTATTGA 3'

BmCHR1 nc 6486557: 5' CCTCTCGCTTTTCCATCGTCT 3'

The cycling conditions were as follows: one cycle of 4 min at 94°C followed by 30 cycles consisting of 30 sec at 94°C, 30 sec at 55°C and 2 min at 72°C. The reaction was completed with a final extension for 10 min at 72°C. The reaction product was diluted 1/20 in deionized water and 1μl of this product was used as a template in a second PCR using RPS12 UTR F and BmCHR1 nc 6486371 5' GTTTTATCGCGCTCGTTCCA 3' as the primers. Cycling conditions consisted of 1 min at 94°C followed by 30 cycles of 30 sec at 94°C, 30 sec at 55°C and 2 min at 72°C. The reaction was completed with a final extension for 10 min at 72°C. The products from the second reaction were analyzed by electrophoresis on a 1% agarose gel. The DNA sequences of the resulting purified amplicons were then determined by a commercial sequencing service (Eurofin Genomics, Louisville, KY).

## Adaptor Mediated PCR assay for off Target Insertion Sites

The DNA sequence of pCHR1-BmGLuc-GFP and its predicted insertion site were examined to identify a restriction enzyme that would cut once in the pCHR1-BmGLuc-GFP cassette to be integrated into the genome but would also and cut outside of the flanking DNA sequence included in pCHR1-BmGLuc-GFP. The restriction enzyme *Nru* 1 was found to match these criteria. Purified DNA samples from individual microfilaria were digested with *Nru* 1 (New England Biolabs) following the manufacturer's instructions. The digest products were purified using the QIAquick PCR purification kit (Qiagen). Purified DNA was recovered in 30 μl of water.

The purified *Nru* 1 restriction fragments were then ligated to an adaptor to facilitate subsequent amplification. Two complementary oligonucleotides were used in this process, with the following sequences:

Nru 1F 5' AGTAGGAAAGTCCCGTAAGG 3'

Nru 1R 5' CCTTACGGGACTTTCCTAC 3'

A total of 5 μl of 10uM Nru 1 F was phosphorylated withT4 Polynucleotide Kinase (New England Biolabs) in a final volume of 50 μl, following the manufacturer's instructions. Upon completion of the reaction, 5 μl of a solution of 10 μM Nru 1R was added to the reaction. The reaction was heated to 94°C for 5min and allowed to cool slowly to 37°C. The annealed phosphorylated adaptor (1 μl) was combined with 30 μl of the purified DNA from above, and the adaptor and DNA ligated in a total volume of 50 μl with T4 DNA ligase (New England Biolabs), following the manufacturer's instructions.

The insertion sites were amplified from the adaptor ligated genomic DNA using a Hemi-nested PCR protocol. Hemi-Nested PCR was performed using Thermo Maxima DreamTaq Hot Start PCR master mix (Cat# K9011). Reactions contained 25 μl Thermo Maxima Dream-Taq Hot Start PCR master mix, 5 μl of adaptor ligated DNA solution, 1μl of 10 μM of RPS12UTR EXT primer (5' CTGTCCACACAATCTGCCCT 3'), 1 μl of 10uM Nru 1R in a total volume of 50 μl. The cycling conditions were as follows: One Cycle of 94°C (4min) followed by 30 cycles 9 °C (30s), 55°C (30s), 72°C (10min). The reaction was completed with a final extension for 10 min at 72°C. The amplification product was diluted 1/20 in water and used as a template in the nested PCR. The nested PCR reaction contained 25 μl Thermo Maxima DreamTaq Hot Start PCR master mix, 1 μl of the diluted first PCR product, 1 μl of 10 μM of RPS12UTR INT primer (5' GACTATTGTTTGTTTATTGTTTGTATTGA 3'), 1 μl of 10 μM Nru 1R in a total of 50 μl. The cycling conditions were as follows: One Cycle at 94°C

(1min), followed by 30 cycles consisting of 94˚C (30s), 55˚C (30s), 72˚C (10min). The reaction was completed with a final extension at 72˚C for 10 min. The products from the second reaction were analyzed by electrophoresis on a 1% agarose gel.

## Results

As a first step in determining if CRISPR could be used to insert genes at a specific position in the *B. malayi* genome, the genomic sequence was analyzed to identify potential insertion sites. We attempted to identify "genomic safe harbors" [20] where integration of the reporter cassette would be unlikely to disrupt the normal function of nearby genes. In identifying such sites, we used the following criteria: 1. The intergenic regions had to be unique in the genome; 2. They had to contain a terminal protospacer adjacent motif (PAM) necessary for recognition by the sgRNA/CAS9 RNP (NGG); and 3. The putative PAM sequence had to be at least 2kb from the nearest predicted coding region. Based on these criteria we identified three candidate insertion sites that would represent potential genomic safe harbors (Table 1). One of these, Chr1scaffold 001:6484000–6485999 (*B. malayi* v4 assembly available in the European Nucleotide Archive database under accession number GCA_000002995.5 [https://www.ebi.ac.uk/ena/data/view/GCA_000002995.5]), was chosen at random to serve as the insertion site.

Chr1scaffold 001:6484000–6485999 was then analyzed to identify the best target location for CRISPR cutting and insertion as described in the Materials and Methods. Based upon this analysis, the sequence corresponding to positions 6485106–6485125 in Chr1 scaffold 001 was identified as the sequence best suited to serve as the sgRNA target. The genomic sequences upstream (6484000–6485105; 1106 nt) and downstream (6485129–6485987; 859 nt) of the sgRNA target site and its 3' PAM motif were then combined with the dual reporter expression cassette in pBACIIBmGLuc-GFP [19]. This resulted in the production of pCHR1-BmGLuc-GFP (Fig 1).

This plasmid contains the upstream (1106 nt) and downstream (859 nt) sequences of the sgRNA recognition site flanking a dual reporter cassette. The cassette contains a modified *Gaussia princips* secreted luciferase (GLuc) under the control of the Bm*hsp70* promoter and Bm*hsp70* 3' UTR, and the green florescent protein (GFP) ORF under the control of the Bm*rps12* promoter and 3' UTR.

An sgRNA was then prepared corresponding to the sequence encoded in positions 6485106–6485125 of Chr1 scaffold 001. To confirm that the sgRNA could specifically direct cutting at the target sequence, the sgRNA was first tested in an *in vitro* cutting assay with a PCR amplicon amplified from *B. malayi* genomic DNA that contained the sgRNA target sequence, as well as the upstream and downstream sequences present in pCHR1-BmGLuc-GFP. The sgRNA, when combined with the amplicon and CAS9 nuclease, cut the amplicon into 1123 and 877 bp fragments, consistent with the predicted sizes of the fragments had the sgRNA directed the CAS9 nuclease to cut the amplicon at the target site (Fig 2). Cutting was not observed in the absence of either the CAS9 nuclease or the sgRNA (Fig 2).

The sgRNA, Cas9 nuclease, and pCHR1-BmGLuc-GFP were then introduced into molting *B. malayi* L3 by lipofection as described in the Materials and Methods. The transgenic parasites

**Table 1. Putative Safe Harbor Sites for CRISPR Insertion of the Reporter Cassette.**

| Location | Upstream UTR (Distance in nt) | Downstream UTR (Distance in nt) |
|---|---|---|
| Chr1scaffold 001:14386500–14387872 | Bm433 (6560) | Bm8071 (13807) |
| Chr1scaffold 001:6484000–6485999 | Bm3846 (9870) | Bm13263 (9871) |
| Chr1scaffold 001:9808670–9810425 | Bm10782 (6368) | Bm5327 (28,626) |

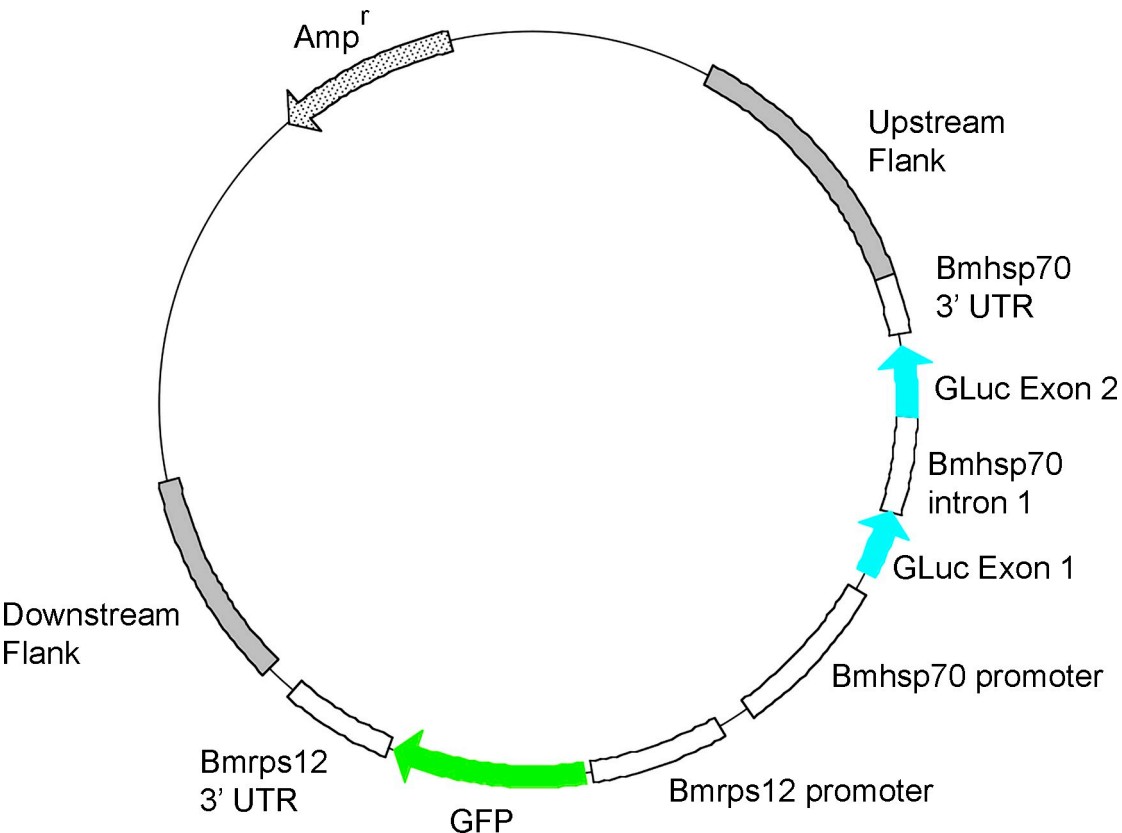

**Fig 1. Map of pCHR1-BmGLuc-GFP: Reporter open reading frames and direction of transcription are indicated by the colored arrows.**

were injected intraperitoneally into two naïve gerbils and allowed to develop to adults and produce F1 progeny microfilariae, as previously described [10]. The microfilariae from the gerbils were isolated, placed in individual wells of 96 well plates, and were assayed for secreted GLuc reporter activity, as previously described [19]. A total of 1440 individual microfilariae (960 from one animal and 480 from the second animal) were screened in this manner. The results of the GLuc assay revealed that the individual microfilaria segregated into two distinct groups. Most of the microfilariae (n = 1394) analyzed secreted low levels of GLuc activity that were not different from those seen from untransfected individual microfilariae (Fig 3). However, secreted GLuc activity detected into the culture medium from 46 microfilariae was 7 to 8-fold higher than for the 1394 parasites in this low activity group (p < 0.001; t test), indicating that these F1 progeny may be carrying the transgenes (Fig 3). This represented a ~3% prevalence of transgenic F1 parasites, which was consistent with the results previously obtained with parasites transfected with the *piggyBac* constructs [10].

DNA from the individual GLuc positive microfilaria was then assayed by hemi-nested PCR for the presence of the transgenic sequences and for their insertion point in the genome. All of the GLuc positive microfilariae (n = 46) produced PCR amplicons consistent with insertion of the transgenic sequences into the expected location in the genome, while no product was detected from randomly selected individuals from the population of 1394 microfilariae that produced activity equivalent to that seen from untransfected microfilariae, or from *B. malayi* genomic DNA prepared from an untransfected female parasite (Fig 4).

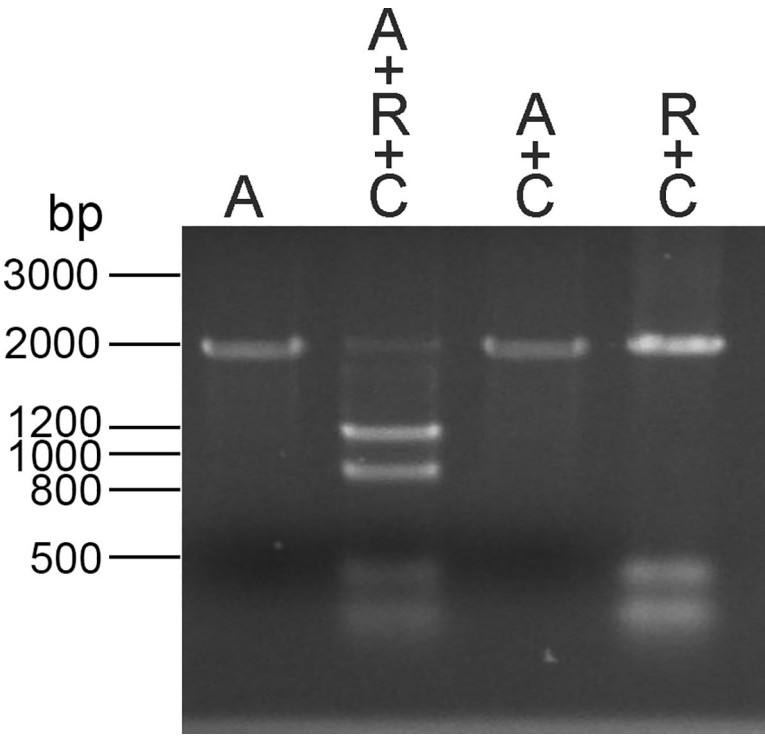

**Fig 2.** *In vitro* digestion of the putative insertion site: An amplicon corresponding to Chr1scaffold 001:6484000–6485999 was produced and digested with the corresponding sgRNA and CAS9 nuclease, as described in Materials and Methods. Lane A = undigested amplicon. Lane A+R+C = amplicon incubated with the sgRNA and CAS9 nuclease. Lane A+C = amplicon incubated with CAS9 nuclease alone. Lane A+R = amplicon incubated with the sgRNA alone.

To further confirm the insertion site, the DNA sequence of the amplicons from 10 randomly selected microfilariae were determined. All matched the sequence predicted for insertion of the transgenic sequences into the site targeted by the sgRNA.

Although all of the transgenic microfilariae contained the insertion at the predicted site, it was considered possible that insertions might have occurred at additional off target sites in the genome in the transgenic parasites. To detect such off-target insertions, DNA from 14 individual transgenic microfilariae was digested with *Nru* 1 and an adaptor was ligated to the ends of the resulting restriction fragments. The adaptor ligated products were then amplified using a hemi-nested PCR assay to identify the insertion sites, as described in the Materials and Methods. All 14 of the DNA samples from the individual transgenic microfilaria produced a single PCR amplicon, demonstrating that they contained a single insertion site, while no product was detected from DNA prepared from 13 non-transgenic microfilariae (Fig 5).

The single band seen in all the transgenic microfilariae was approximately 2400 bp, consistent with the amplicon predicted to be produced from insertion into the predicted site (2374 bp). DNA sequence analysis of the amplicons confirmed that they were derived from the expected insertion site.

## Discussion

The results presented demonstrate that CRISPR can be used to direct insertions into defined locations in the genome of *B. malayi*. Previously, we have developed a toolkit for integrative stable transfection of *B. malayi* that utilized the *piggyBac* transposon-based system. The

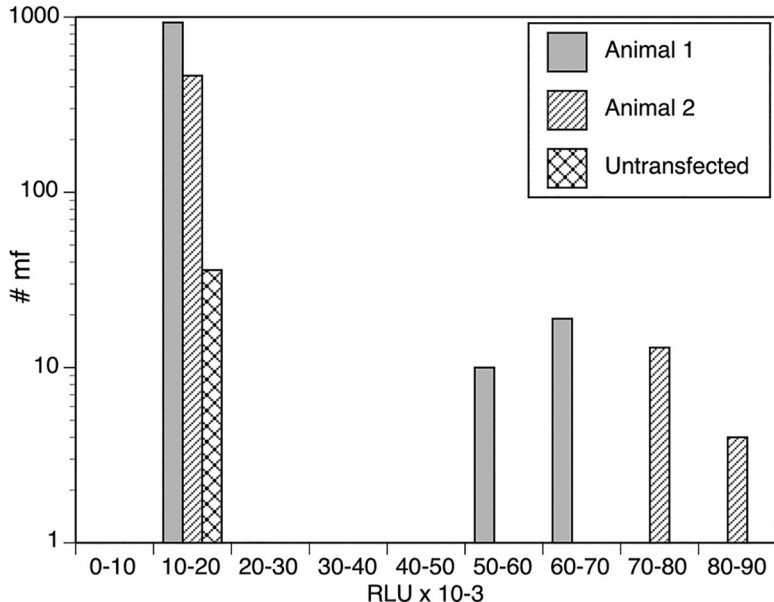

**Fig 3.** Gaussia luciferase activity secreted by individual F1 microfilaria: Microfilariae were collected from two gerbils that were independently infected with transfected molting L3 and cultured overnight, and one gerbil infected with untransfected L3. The media for the cultured individual microfilaria were assayed for Gaussia luciferase activity, as described in Materials and Methods. The number of microfilariae producing luciferase activity corresponding to each interval on the X axis are represented by the bars. The Y axis is presented using a logarithmic scale for clarity. #mf = number of mf secreting the indicated range of Gaussia luciferase activity into the culture medium. RLU = Relative light units.

*piggyBac* system results in integration into loci with the sequence TTAA, which are widely distributed in the genome. While this semi-random mechanism of insertion can be quite useful in some instances (for example, conducting saturation mutagenesis of the *Plasmodium falciparum* genome [21]), the ability to direct insertions into specific locations in the genome confers

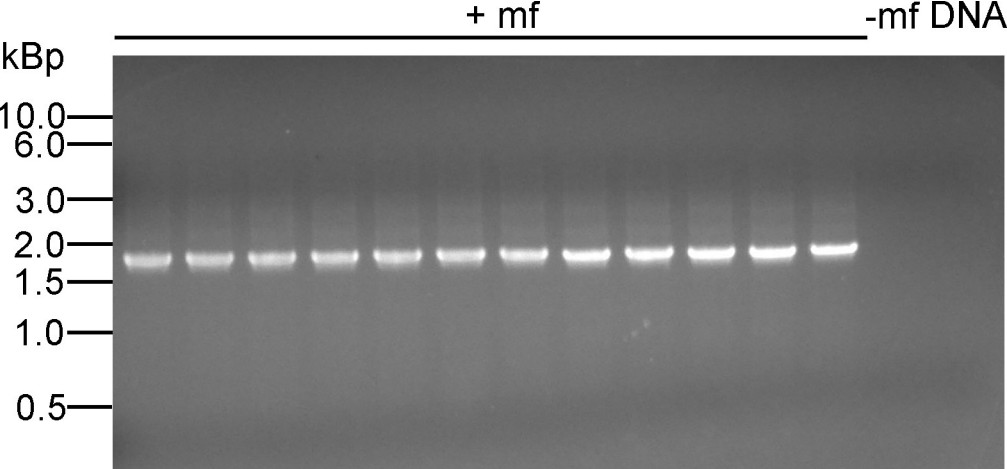

**Fig 4.** Hemi-nested PCR analysis of insertion sites in individual microfilaria secreting Gaussia luciferase: Results shown are 12 representatives of those obtained from all 46 luciferase positive microfilariae recovered. Lanes labeled mf+ = DNA prepared from individual microfilaria positive for luciferase activity. Lane mf- = DNA prepared from an individual microfilaria recovered from the animals that did not secrete luciferase. Lane DNA = genomic DNA prepared from untransfected adult female *B. malayi* DNA.

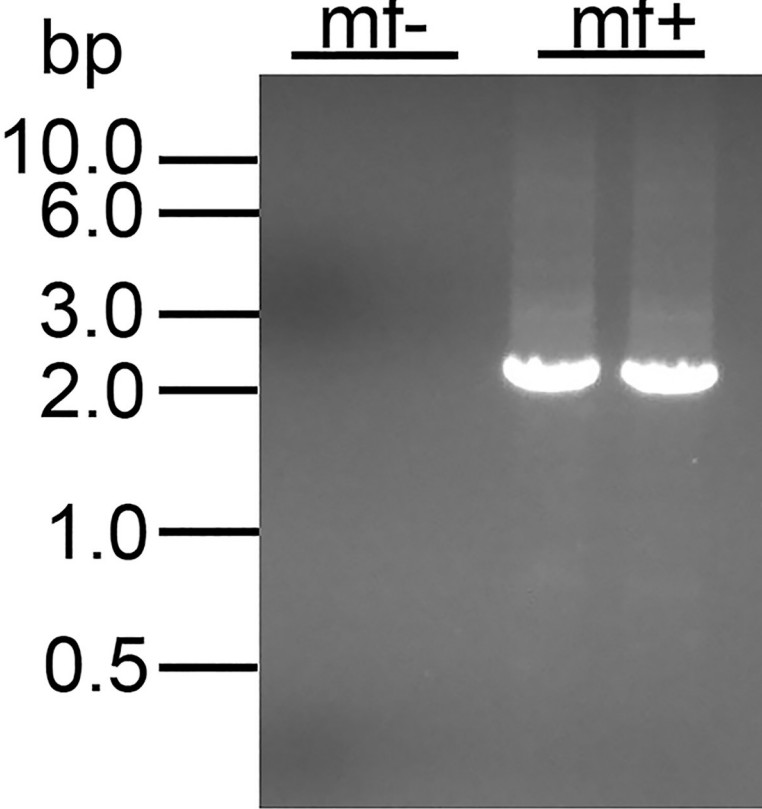

**Fig 5.** Adaptor mediated hemi-nested PCR assay for off target insertion sites: Results shown are representatives from 14 individual transgenic microfilariae and 13 non-transgenic microfilariae, all of which gave results identical to those shown here. Lanes labeled mf- = non-transgenic microfilaria and those labeled mf+ = transgenic microfilaria.

several advantages for the production of knock-in mutations. For example, using CRISPR, it should be possible to insert constructs carrying various promoters driving the expression of a reporter gene into a single designated location in the genome. By doing so, one can eliminate any position effects that might result from a more random insertion into different locations in the genome. Furthermore, if all insertions are located at the same position in the genome, all individual transgenic parasites will carry exactly the same insertion. This will make the process of crossing parasites to obtain a homozygous parasite line less complicated to perform.

Another important potential application of CRISPR in *B. malayi* will be precise editing of sequences in the genome. Thus, it should be possible to knock out specific genes or to create conditional mutants, permitting one to carry out functional genomic studies on this parasite. It should also be possible to introduce alleles found to be associated with drug resistance into susceptible parasites, allowing one to directly test the hypothesis that a particular mutation is responsible for the development of resistance. However, in this study, we have not yet demonstrated that CRISPR will be capable of such precise editing in *B. malayi*. Studies addressing this question are currently underway.

While the results presented here demonstrate that CRISPR can be used in *B. malayi*, some obstacles remain to be overcome before this tool can be used to routinely and efficiently develop transgenic parasite lines. First, it will be necessary to develop a way to efficiently screen for transgenic F1 microfilariae or L3. In the current study, we found that roughly 3% of the F1

microfilariae produced from adult parasites transfected as L3 were transgenic. This percentage is very similar to what was obtained using the *piggyBac* system [10] and so may be a characteristic of the system rather than specific to the CRISPR constructs used here. Given that only 10–15% of the L3 introduced into a gerbil survive to produce adult parasites [22] it seems that a minimum of 30–40 L3 are required to reliably establish a patent infection. If 3% of the F1 parasites are transgenic, this will require screening 1000–2000 F1 progeny to obtain sufficient transgenic parasites to maintain the transgenic line. Manually sorting and screening this number of parasites is quite labor intensive. Thus, more efficient ways to isolate and screen the F1 parasites will be needed if this technology is to be used routinely. Alternatively, the methods used for CRISPR mediated modification might be optimized to produce a larger percentage of transgenic F1 offspring in *B. malayi*. For example, in the work reported here, we have used upstream and downstream arms that were chosen to direct long-range homologous recombination, as the dual reporter gene cassette we inserted was quite large. The sequences we used (1106 nt and 859 nt) were within the range reported to be optimal for CRISPR mediated long range homologous recombination in the free living nematode *Caenorhabditis elegans* (500-1500nt) [23, 24], but larger or smaller arms may prove to be more efficient in directing CRISPR-mediated long range recombination in *B. malayi*. It is also possible that smaller modifications involving gene editing or short range homologous recombination may prove to be more efficient than long range homologous recombination. CRISPR mediated short range homologous recombination has been reported to be more efficient than long range homologous recombination in *C. elegans* [24]. In *Strongyloides stercoralis*, roughly 17% of the F1 progeny in a study using CRISPR mediated short range homologous recombination were transgenic [15]. Other modifications may also improve the efficiency of CRISPR in *B. malayi*. For example, in *Strongyloides stercoralis*, co-transfection with a plasmid expressing the CAS9 nuclease under the control of an endogenous promoter was found to be more efficient in generating transgenic F1 progeny than co-transfection with CAS9 nuclease–sgRNA RNPs [15].

Finally, it is likely that many of the phenotypes of interest (e.g. knockouts of specific parasite genes) will likely be recessive, meaning that they will only manifest themselves in parasites homozygous for the mutant of interest. Thus, it will be necessary to develop ways to genotype the parasites without adversely affecting their ability to survive and develop in the animal host. Work addressing all of these obstacles is currently underway.

## Acknowledgments

We are extremely grateful to the NIH/NIAID Filariasis Research Reagent Resource Center (FR3; www.filariasiscenter.org) for providing us with the parasites used in this study. We would also like to thank Ms. Kristi Miley for technical assistance.

## Author Contributions

**Conceptualization:** Canhui Liu, Alexandra Grote, Elodie Ghedin, Thomas R. Unnasch.

**Data curation:** Elodie Ghedin.

**Formal analysis:** Canhui Liu, Alexandra Grote, Elodie Ghedin, Thomas R. Unnasch.

**Funding acquisition:** Thomas R. Unnasch.

**Investigation:** Elodie Ghedin.

**Methodology:** Canhui Liu, Thomas R. Unnasch.

**Project administration:** Elodie Ghedin, Thomas R. Unnasch.

**Supervision:** Thomas R. Unnasch.

**Writing – original draft:** Alexandra Grote, Elodie Ghedin, Thomas R. Unnasch.

**Writing – review & editing:** Alexandra Grote, Elodie Ghedin, Thomas R. Unnasch.

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
