## [Decision Letter · Decision Letter 0]

2 Jun 2020

Dear Dr. Unnasch,

Thank you very much for submitting your manuscript "CRISPR-mediated Transfection of Brugia malayi" for consideration at PLOS Neglected Tropical Diseases. As with all papers reviewed by the journal, your manuscript was reviewed by members of the editorial board and by several independent reviewers. In light of the reviews (below this email), we would like to invite the resubmission of a significantly-revised version that takes into account the reviewers' comments. 

We cannot make any decision about publication until we have seen the revised manuscript and your response to the reviewers' comments. Your revised manuscript is also likely to be sent to reviewers for further evaluation.

Sincerely,

Sabine Specht

Associate Editor

Timothy Geary

Deputy Editor

Reviewer's Responses to Questions

**Key Review Criteria Required for Acceptance?**

**Methods**

-Are the objectives of the study clearly articulated with a clear testable hypothesis stated?

-Is the study design appropriate to address the stated objectives?

-Is the population clearly described and appropriate for the hypothesis being tested?

-Is the sample size sufficient to ensure adequate power to address the hypothesis being tested?

-Were correct statistical analysis used to support conclusions?

-Are there concerns about ethical or regulatory requirements being met?

Reviewer #1: The methods in this study are clearly articulated. This paper reports a very significant technical achievement in genome modification of B. malayi and is not explicitly hypothesis driven. This is appropriate for such a report. Sample sizes are sufficient to support the claims of the paper, and statistical analyses were used. An additional control involving microfilariae from non-transgenic adults would clarify the status of worms exhibiting an apparent low-level of GLuc expression in Figure 3. There are no concerns about ethical or regulatory issues.

Reviewer #2: Excellent. Only minor corrections to the text.

**Results**

-Does the analysis presented match the analysis plan?

-Are the results clearly and completely presented?

-Are the figures (Tables, Images) of sufficient quality for clarity?

Reviewer #1: The analysis presented matches the plan

The issue of the status of mff apparently expressing a low level of GLuc expression, alluded to in the summary needs to be addressed.

Figures and tables are OK, but authors should consider revising Fig. 2 along lines suggested in Substantive Point 12 in the summary.

Reviewer #2: Very clear. Only minor corrections to the text.

**Conclusions**

-Are the conclusions supported by the data presented?

-Are the limitations of analysis clearly described?

-Do the authors discuss how these data can be helpful to advance our understanding of the topic under study?

-Is public health relevance addressed?

Reviewer #1: Conclusions are supported by the data, and limitations of the system are clearly addressed. 

The discussion would be greatly improved if it placed the current findings on B. malayi in the broader context of CRISPR/Cas9 mutagenesis in other parasitic helminths.

Public health relevance of the reported findings is addressed

Reviewer #2: Straight to the point! Only minor corrections to the text.

**Editorial and Data Presentation Modifications?**

Reviewer #1: Seven suggestions in this area are presented under MINOR POINTS OF GRAMMAR AND USAGE in the summary,

Reviewer #2: Only very minor corrections/modifications. These are indicated in the attached word document. I recommend rapid publication.

**Summary and General Comments**

Reviewer #1: Review of Manuscript No. PNTD-D-20-00588 by Liu, Unnasch and others entitled “CRISPR-mediated Transfection of Brugia malayi”. The authors, who have previously achieved both transient and integrative transgenesis in B. malayi, report here the integration of a single copy of a reporter transgene into a large intergenic region in the genome of this parasite by CRISPR/Cas9. B. malayi, a human pathogen in its own right, is regarded widely as a model for the other parasites that cause lymphatic filariasis. This is an exceedingly important finding for the field and marks an important step forward in what has been a long record of success by this group of authors in devising functional genomic methods for the lymphatic dwelling filariae. These achievements are remarkable considering the difficult technical hurdles conferred by the fact that there are no free-living stages of B. malayi or any other filaria, and that both a mosquito and a mammalian host are required for the parasite to complete its life cycle. 

This paper has many strengths. It reports the first proof of principle for CRISPR/Cas9 genome modification in these very important pathogens. In doing so, the authors have succeeded in precisely targeting a single-copy reporter transgene integration to what could be characterized as a genomic safe harbor locus in the B. malayi genome. They take pains to demonstrate that there were no off-target integrations produced by their system. In general, the authors do a good job of discussing their findings in the context of previous work on transgenesis in B. malayi.

The paper also has some substantive deficiencies and areas for improvement, which are enumerated below. Chief among these is that the reported findings, important as they are, are discussed rather narrowly in the context of previous work on B. malayi, rather than in a wider context of contemporary efforts to achieve CRISPR/Cas9 genome modification in other parasitic helminths, both nematodes and trematodes. The paper would be greatly improved by placing the new findings in B. malayi in this broader context. 

Another technical area of the paper that needs to be addressed and clarified is the status of the 1394 F1 microfilariae that appear in Figure 3 to be displaying a low level of GLuc activity, but are subsequently characterized as non-transgenic parasites. If the authors regard the GLuc activity seen in these worms to be a “background” signal, they should explicitly state that and provide an explanation for the origin of this low-level signal. Ideally, the inclusion of control microfilariae from a gerbil infected with non-transgenic L3 in this figure would resolve this.

Another substantive clarification that would be significant for the paper would be to provide the length(s) in bp of the homology arms in the transgene construct (ie stretches of sequence homologous to sequences flanking the predicted double-stranded break). The lengths of these homology arms might bear on the 3% frequency of integration the authors observe. 

Other points and recommendations for substantive improvements are enumerated below. Following that, a few minor points of grammar and usage are listed. 

SUBSTANTIVE ISSUES AND SUGGESTIONS

1. Line 96: Is it worth mentioning in this area the potential for unintended insertional mutagenesis due to widespread piggyBac integrations?

2. Line 146: It would be informative to insert the length in bp of the amplicon here. There's evidence that longer homology arms give higher efficiency of integration.

3. Line 162: Again, it would be good to give the length of this homology arm in bp.

4. Line 315: Sites such as these have been characterized elsewhere as "genomic safe harbors". Would it be appropriate/useful to do that here?

5. Line 336: Again, need to indicate the lengths of the flanking homology arms in bp.

6. Line 365: Then to what do you attribute the apparent low level of GLuc activity seen in the 1394 parasites? Is it simply "background"? This is a very important clarification for the paper. A control comprising microfilariae from a gerbil infected with non-transgenic L3 would be important in resolving this.

7. Line 377: How about rates of CRISPR/Cas9 mutation/editing in other parasitic helminths (nematodes or flatworms)? The paper would be greatly improved by a discussion of the present findings in the larger context of CRISPR/Cas9 genome modification in other parasitic helminths.

8. Line 382: Are these the 1394 mff referred to in line 365? It'd be good to clarify whether you consider those to be GLuc negative worms. 

9. Line 428: Wouldn't single-copy transgene integrations by CRISPR/Cas9 also eliminate the confounding effects of transgene over expression from the variable but potentially large numbers of piggyBac insertions?

10. Line 447: Some mention of mutation frequencies produced in other parasitic helminths by CRISPR/Cas9 would enhance this part of the discussion. Also, knowing the length of the homology arms used in this study might reflect on this. There may be evidence that the length of these can enhance frequency of integration.

11. Line 450: C. elegans workers have devised a fluorescence activated sorting system for transgenic worms, and this has been "piloted" in transgenic parasitic nematodes. Would that be worth adding to the discussion?

12. Figure 2 and caption (lines 350-354): Recommend substituting abbreviated labels for these code numbers, making it easier to interpret the gel without referring to text or caption to decode.

MINOR POINTS OF GRAMMAR AND USAGE

1. Line 63: suggest Insert "agents" after "causative".

2. Line 219: Consider using the singular "medium" here and following where appropriate. Should be singular in the next sentence too.

3. Line 396: Make it "microfilariae"....plural, right?

4. Line 397: Suggest adding "considered" before "possible".

5. Line 399: Make it "was digested" (subject verb agreement)

6. Line 402: Shouldn't it be "microfilariae"...plural?

7. Line 429: Suggest "...will be precise editing of genome sequences."

Reviewer #2: This is definitely the most important paper in filarial biology that I have read this year. It is an excellent follow-up to their recent paper on successful transfection of Brugia malayi. Along with transfection, the application of CRISPR technology to filarial parasites will prove to be one of the most significant technological advances needed to elucidate the biology of these difficult organisms. The paper is well written and clear. I have made a few suggestions for clarity and caught a few grammar issues/typos, but no significant modifications are necessary and I recommend quick publication.

PLOS authors have the option to publish the peer review history of their article (what does this mean?). If published, this will include your full peer review and any attached files.

Reviewer #1: No

Reviewer #2: No
---

## [Decision Letter · Decision Letter 1]

21 Jul 2020

Dear Dr. Unnasch,

We are pleased to inform you that your manuscript 'CRISPR-mediated Transfection of Brugia malayi' has been provisionally accepted for publication in PLOS Neglected Tropical Diseases.

Best regards,

Sabine Specht

Associate Editor

Timothy Geary

Deputy Editor

Reviewer's Responses to Questions

**Key Review Criteria Required for Acceptance?**

**Methods**

-Are the objectives of the study clearly articulated with a clear testable hypothesis stated?

-Is the study design appropriate to address the stated objectives?

-Is the population clearly described and appropriate for the hypothesis being tested?

-Is the sample size sufficient to ensure adequate power to address the hypothesis being tested?

-Were correct statistical analysis used to support conclusions?

-Are there concerns about ethical or regulatory requirements being met?

Reviewer #1: Objectives of the study are clearly articulated and study design is appropriate for the stated objectives. This is a report of new methodology for Brugia and not a hypothesis driven study, which is entirely appropriate. The study population is clearly described and sample sizes are sufficient. Analysis and interpretation of data are satisfactory. Ethical issues are satisfactorily addressed.

Reviewer #2: Nothing additional to add to my earlier review. The paper is excellent and meets all criteria for publication.

**Results**

-Does the analysis presented match the analysis plan?

-Are the results clearly and completely presented?

-Are the figures (Tables, Images) of sufficient quality for clarity?

Reviewer #1: Yes, the analysis matches the analysis plan and results in the revised manuscript are now clearly and completely presented. Figures and tables are of sufficient quality for clarity.

Reviewer #2: Nothing to add to my earlier review.

**Conclusions**

-Are the conclusions supported by the data presented?

-Are the limitations of analysis clearly described?

-Do the authors discuss how these data can be helpful to advance our understanding of the topic under study?

-Is public health relevance addressed?

Reviewer #1: Conclusions are certainly supported by the data presented, and limitations of the analysis are described. The impact of these data and public health relevance are clearly presented in the manuscript narrative.

Reviewer #2: Nothing to add to my earlier review.

**Editorial and Data Presentation Modifications?**

Reviewer #1: The revised manuscript incorporates all crucial points raised by this reviewer about the original submission. I support the authors' decision not to discuss the possible application of fluorescence activated parasite sorting.

Reviewer #2: No modifications suggested.

**Summary and General Comments**

Reviewer #1: This is a very important paper, reporting the first achievement of targeted mutagenesis in a filarial worm via CRISPR/Cas9. It draws significantly on the authors' previous achievements in transgenesis and transgene integration into the chromosomes of Brugia. The authors have done a very conscientious job of revising the manuscript and have incorporated virtually all suggestions by this reviewer following the initial review.

Reviewer #2: As described in my original review, this is a very important paper with implications to revolutionize work on filarial parasites. The changes made to the paper since my last review are all acceptable and slightly improve the quality of the paper.

PLOS authors have the option to publish the peer review history of their article (what does this mean?). If published, this will include your full peer review and any attached files.

Reviewer #1: No

Reviewer #2: No

---

## [Editor Report · Acceptance letter]

25 Aug 2020

Dear Dr. Unnasch,

We are delighted to inform you that your manuscript, "CRISPR-mediated Transfection of *Brugia malayi*," has been formally accepted for publication in PLOS Neglected Tropical Diseases.

Best regards,

Shaden Kamhawi

co-Editor-in-Chief

Paul Brindley

co-Editor-in-Chief
